# Strategies of Influenza A Virus to Ensure the Translation of Viral mRNAs

**DOI:** 10.3390/pathogens11121521

**Published:** 2022-12-12

**Authors:** Hui-Chun Li, Chee-Hing Yang, Shih-Yen Lo

**Affiliations:** 1Department of Biochemistry, Tzu Chi University, Hualien 97004, Taiwan; 2Department of Laboratory Medicine and Biotechnology, Tzu Chi University, Hualien 97004, Taiwan; 3Department of Laboratory Medicine, Buddhist Tzu Chi General Hospital, Hualien 97004, Taiwan

**Keywords:** influenza A virus, host shutoff, innate immune responses, selective translation of viral mRNAs, NS1, PA-X, PB1-F2, PABP1, IFIT2

## Abstract

Viruses are obligatorily intracellular pathogens. To generate progeny virus particles, influenza A viruses (IAVs) have to divert the cellular machinery to ensure sufficient translation of viral mRNAs. To this end, several strategies have been exploited by IAVs, such as host gene shutoff, suppression of host innate immune responses, and selective translation of viral mRNAs. Various IAV proteins are responsible for host gene shutoff, e.g., NS1, PA-X, and RdRp, through inhibition of cellular gene transcription, suppression of cellular RNA processing, degradation of cellular RNAs, and blockage of cellular mRNA export from the nucleus. Host shutoff should suppress the innate immune responses and also increase the translation of viral mRNAs indirectly due to the reduced competition from cellular mRNAs for cellular translational machinery. However, many other mechanisms are also responsible for the suppression of innate immune responses by IAV, such as prevention of the detection of the viral RNAs by the RLRs, inhibition of the activities of proteins involved in signaling events of interferon production, and inhibition of the activities of interferon-stimulated genes, mainly through viral NS1, PB1-F2, and PA-X proteins. IAV mRNAs may be selectively translated in favor of cellular mRNAs through interacting with viral and/or cellular proteins, such as NS1, PABPI, and/or IFIT2, in the 5′-UTR of viral mRNAs. This review briefly summarizes the strategies utilized by IAVs to ensure sufficient translation of viral mRNAs focusing on recent developments.

## 1. Introduction

Influenza A viruses (IAV) are highly contagious human pathogens that cause seasonal endemics and sometimes pandemics. IAV infections remain a global problem since available vaccines and anti-viral agents are not always effective to combat them. The genome of IAV contains 8 segments of single-stranded, negative-sense RNAs, which encode at least 17 proteins [1], including polymerase basic 2 (PB2) from segment 1, polymerase basic 1 (PB1), PB1 frame 2 (PB1-F2), and PB1-N40 from segment 2 [2], polymerase acidic (PA), PA-X, PA-N155, and PA-N182 from segment 3 [3,4], hemagglutinin (HA) from segment 4 [5], nucleoprotein (NP) from segment 5 [6], neuraminidase (NA) from segment 6 [7], matrix proteins 1 (M1), 2 (M2), and M42 from segment 7 [8,9,10], and non-structural proteins 1 (NS1), 2 (NS2, nuclear export protein), and 3 (NS3) from segment 8 [11,12] (Table 1). The IAV particles are enveloped with two abundant glycoproteins, HA and NA, and a few M2 proteins. The M1 proteins localize beneath the viral envelope and interact with both viral membrane and RNA to scaffold viral particles. There are eight negative-sense RNA segments in viral genome; each segment is coated with multiple copies of NP and associated with RNA-dependent RNA polymerase complexes (RdRp, composed of PA, PB1, and PB2) to form viral ribonucleoprotein complexes (vRNPs) (for a review, see [13]). IAVs can be subtyped by HA and NA glycoproteins. At present, there are at least 18 different HA subtypes and 11 different NA subtypes found in the wild. The affected hosts include humans, birds, and pigs [14].

The infection of IAV begins with binding of HA to sialic acid on the host membrane receptors, which triggers the receptor-mediated endocytosis. The virus is then engulfed into endosome, where the low pH environment initiates a conformational change of HA that mediates membrane fusion to release vRNPs in the cytosol [15,16]. Released vRNPs translocate into the nucleus, where viral mRNAs (i.e., transcription) and anti-genomic RNAs (i.e., replication) are synthesized using the viral genomic RNAs as templates [17,18]. The RdRp is responsible for replication and transcription of the eight viral RNA segments. The RdRp generates viral mRNAs using short-capped primers derived from cellular transcripts of RNA polymerase II (RNAP II) by a unique ‘cap-snatching’ mechanism: the PB2 subunit binds the 5′ cap of cellular RNA transcripts [19], which are subsequently cleaved after 10–13 nucleotides by the viral endonuclease, PA [20]. The 3′-poly(A) tails of IAV mRNAs are synthesized by reiterative copying of the U track near the 5′ end of the viral genomic RNAs [21]. Viral mRNAs are then exported to the cytosol for translation. Anti-genomic RNAs serve as templates for the generation of more viral genomic RNAs. Each viral RNA segment is encapsulated by PA, PB1, PB2, and NP proteins to form the vRNP. Progeny vRNPs are exported to the cytosol with the help of M1 and NS2 proteins [22]. Then, the vRNPs, together with HA, NA, M1, and M2 proteins, assemble to form the viral particles beneath the plasma membrane [23]. Finally, the new virions are budded and released from cells facilitated by NA-mediated cleavage of sialic acid (for a review, see [24]).

To generate progeny virions successfully, IAVs have to divert host cellular apparatuses to ensure sufficient translation of viral mRNAs. Several strategies utilized by IAV have been characterized, such as host gene shutoff, suppression of host innate immune response, and selective translation of viral mRNAs.

**Table 1 pathogens-11-01521-t001:** Summary of major functions of IAV individual proteins.

Gene Segment (nucleotides *)	Protein	Functions	References
1 (2313 nt)	PB2	Recognizes cap structure	[19]
2 (2302 nt)	PB1	Elongation of RNA synthesis	[25]
	PB1-F2	Inhibition of host immune responses	[26,27]
	PB1-N40	Supports the RdRp activity	[2]
3 (2206 nt)	PA	Endonuclease activity for cap-snatching	[20]
	PA-X	Host shutoff	[28,29]
	PA-N155	Supports virus replication	[3]
	PA-N182	Supports virus replication	[3]
4 (1746 nt)	HA	Mediates virus attachment to cells	[5]
5 (1537 nt)	NP	Encapsulates viral genomes	[6,17]
6 (1370 nt)	NA	Facilitates virus release	[30]
7 (996 nt)	M1	Virion assembly	[8]
	M2	Ion channel; virus budding	[9,23]
	M42	Supports virus replication	[10]
8 (854 nt)	NS1	Inhibition of host immune responses	[31]
	NS2	Nuclear export of vRNPs	[11,22]
	NS3	Host specificity	[12]

* strain: A/WSN/1933 TS61(H1N1) [https://www.ncbi.nlm.nih.gov/nuccore/?term=txid370129[Organism:noexp] accessed on 12 August 2022].

## 2. Host Gene Shutoff

All viruses use cellular translational machinery to decode viral mRNAs. Viruses have to re-direct the cellular machinery to ensure sufficient translation of viral mRNAs. Host shutoff is a common strategy used by many viruses to repress cellular gene expression and promote viral protein synthesis for replication [32]. For example, infections with alphaherpesviruses, gammaherpesviruses, or betacoronaviruses result in cellular mRNA degradation due to translatable cellular mRNAs cleaved by viral proteins (e.g., virion host shutoff protein of herpes simplex virus, SOX protein of Kaposi’s sarcoma-associated herpesvirus, and nsp1 of coronaviruses) and then degraded completely by cellular Xrn1 [33]. IAV infection results in lower level of cellular proteins synthesis and accumulation of viral proteins in virus-infected chicken embryo fibroblasts cells [34]. IAV deploys multiple mechanisms to suppress the cellular gene expression (i.e., host shutoff) mainly through several viral proteins: NS1, PA-X, and RdRp. The underlining mechanisms include inhibition of cellular gene transcription, suppression of cellular RNA processing, degradation of cellular RNAs, blockage of cellular mRNA export from the nucleus, and selective translation of viral mRNAs [35].

### 2.1. Inhibition of Cellular Gene Transcription ((1) in Figure 1)

#### 2.1.1. Degradation of RNA Polymerase II by Viral RdRp

IAV infection independent of the viral strains results in reduction of cellular RNA levels. Further studies indicated that IAV infection causes specific degradation of the largest subunit of RNAP II, which decreases cellular RNA synthesis [36]. Due to the cap-snatching process, transcription of IAV by viral RdRp depends on cellular RNAP II transcription. Not surprisingly, the viral RdRp complex was found to interact with the large subunit of RNAP II and accumulate at RNAP II transcription sites [37]. Moreover, the expression of viral RdRp alone was sufficient to cause the degradation of RNAP II and also inhibit RNAP II transcription. The PA and PB2 subunits of viral RdRp were identified to be responsible for the degradation of RNAP II from the studies of re-assortant viruses [38]. Increased ubiquitination of RNAP II was detected in both IAV-infected 293T cells and 293T cells expressing viral RdRp, suggesting that the ubiquitin–proteasome system is involved in the degradation of RNAP II [39,40]. Recently, using mammalian native elongating transcript sequencing to examine RNAP II behavior during IAV infection, a study showed that viral infection resulted in the reduction of RNAP II gene occupancy downstream of transcription start sites. Thus, cellular transcription elongation is globally inhibited. In addition, the low level of RNAP II occupancy continues after the 3′ transcription termination and polyadenylation sites, indicating that transcription termination is also impaired [41]. Thus, IAV RdRp plays a role in host shutoff [42], which also contributes to viral virulence [43].

**Figure 1 pathogens-11-01521-f001:**
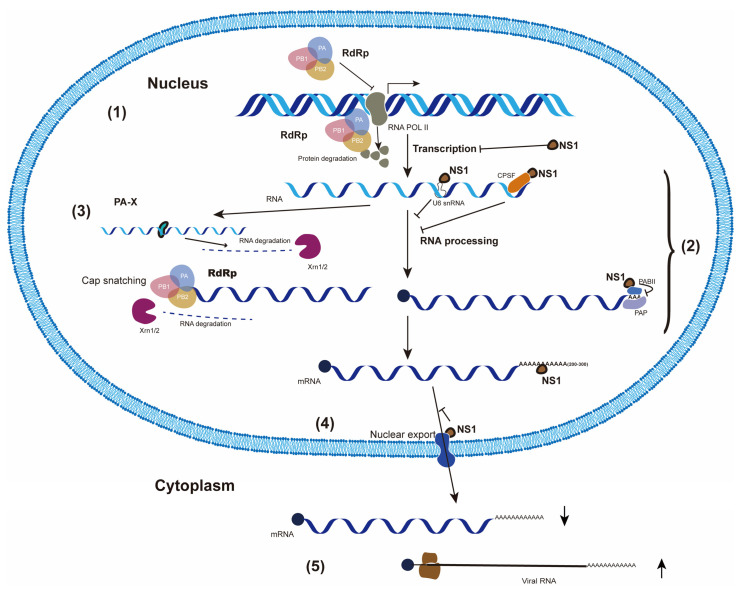
Various mechanisms utilized by IAV to shut off host gene expression. (1) IAV degrades cellular RNAP II by viral RdRp and/or inhibits cellular RNA transcription directly by NS1 to suppress cellular gene transcription; (2) IAV prevents the cellular RNA processing through cap-snatching by viral RdRp and/or NS1 binding to cellular factors involved in the processing of pre-mRNAs, e.g., CPSFs, PABII, U6 snRNA; (3) IAV PA-X degrades cellular RNAs; (4) IAV NS1 blocks the cellular mRNA export from nucleus; (5) IAV diverts the translational machinery to selectively translate viral mRNAs over cellular mRNAs.

#### 2.1.2. NS1 Inhibits Cellular RNA Transcription Directly

NS1 protein is translated from a continuous primary transcript of genome segment 8 and is highly expressed in IAV-infected cells. NS1 is typically a protein of 230 amino acids (a.a.), but often found with varied length (219 to 237 a.a.) in different strains. The IAV NS1 protein contains at least four domains: RNA-binding domain (RBD), linker (L) region, effector domain (ED), and the C-terminal disordered tail (CTT). The N-terminal RBD (the first 73 a.a.) is important for the binding to dsRNA and also for its dimerization. Located in the RBD, residues 35–41 are the nuclear localization signal (NLS). The flexible region of 10–15 a.a. links RBD and ED. There are dozens of cellular factors that interact with NS1 ED (a.a. 88–202) to exert various functions, although not all interactions occur in all IAV strains. The NS1 proteins of most IAV strains contain a nuclear export signal (NES, a.a. 137–146) that results in NS1 protein localization in both the nucleus and the cytosol. NS1 from some IAV strains has a PDZ-binding motif, which affects virus pathogenesis, in CTT (11–33 a.a in length). The NS1 proteins from most IAV strains have a second NLS at its C-terminal end (a.a. 216–221) (for a review, see [31]).

The ARSK (a.a. 226–229) sequence at the C-terminus of the NS1 protein of IAV H3N2 has a similar function as the first four amino acids (ARTK) of histone H3. Analogous to the histone H3K4 methylation, the lysine within the NS1 ARSK sequence is also methylated. The methylated NS1 then interacts with the human PAF1 transcription elongation complex (hPAF1C) and reduces the expression of many cellular genes, including anti-viral genes [44].

A recent study using nascently transcribed RNA analysis in MDCK cells found that functional inducible NS1 inhibited genes driven by RNAP II and RNAP I promoters, but not by non-eukaryotic T7 polymerase. Although NS1 proteins are associated with nuclear chromatin, they do not co-localize with genomic DNA as shown by super-resolution microscopy. NS1 failed to inhibit the expression of reporter genes if artificially tethered to the cytoskeleton. This study also showed that a NS1 without NLS is able to inhibit gene expression as effectively as wild-type NS1, as long as an artificial NLS relocates NS1 to nuclear speckles. This result suggests that the presence of NS1 in the specific structures of the nucleus is essential for host shutoff. Thus, IAV NS1 directly inhibits cellular RNA transcription [45].

### 2.2. Suppression of Cellular RNA Processing ((2) in Figure 1)

#### 2.2.1. Cap-Snatching

IAV lacks an enzyme to add m7G (cap) to the 5′-end of its RNAs. Thus, the viral RdRp uses a unique ‘cap-snatching’ mechanism to get short-capped primers to synthesize viral mRNAs. After cleavage by PA, the un-capped cellular RNAP II transcripts may be prematurely terminated and/or degraded by the cellular exonuclease Xrn1/2, which was thought to be the reason for host shutoff. However, using a non-biased technique (i.e., CapSeq), cellular non-coding RNAs, particularly U1 and U2, were found to be the preferred cap-snatching source over mRNAs or pre-mRNAs. Moreover, promoter-associated capped small RNAs are highly snatched by IAV [46]. Therefore, cap-snatching may not be the main factor responsible for host gene shutoff but may contribute to gene shutoff by altering cellular transcription.

#### 2.2.2. NS1 Interacts with Cellular pre-mRNA Processing Factors

IAV NS1 protein causes host shutoff through interacting with cellular proteins responsible for pre-mRNA processing. The ED of NS1 protein was found to functionally interact with the cleavage and polyadenylation specificity factor 30 (CPSF30), a 30 kDa subunit of the CPSF complex which recognizes the polyadenylation signals at the 3′ end of cellular RNAs, cleaves the RNA, and recruits poly(A) polymerase to add the poly(A) tail [47]. Interaction of NS1 protein with CPSF30 prevents CPSF30 from binding to cellular pre-mRNAs and thus inhibits the 3′-end processing of all cellular pre-mRNAs, which in turn reduces the cellular mRNA level in the cytosol. The host shutoff of NS1 has no effect on the viral mRNAs because the poly(A) tails of viral mRNAs are synthesized by viral RdRp to reiterative copy the U track near the 5′ end of the viral genomic RNA templates [21]. Further studies suggested that the viral RdRp is an integral component of the CPSF30–NS1 protein complex in infected 293T cells. Moreover, PA and NP proteins, but not PB1 and PB2 proteins, are required for stabilizing the CPSF30–NS1 complex [48].

Excessive production of viral glycoproteins (HA and/or NA) during IAV infection could induce endoplasmic reticulum (ER) stress. To balance this, the infected A549 cells provide more ER resident chaperones and reduce translation. Through interfering with CPSF30, NS1 has been demonstrated to suppress ER stress response factors, such as XBP1, to antagonize ER stress induction and benefit viral production [49].

In addition to CPSF30, CPSF subunit 2 and 7 (CPSF2 and CPSF7), which are involved in pre-mRNA polyadenylation machinery, were also demonstrated to interact with NS1 protein of H7N9 strain using co-immunoprecipitation and immunoblotting assays [50].

The NS1 protein ED region interacts with and suppresses the poly(A)-binding protein II (PABII) of the cellular pre-mRNA 3′-end processing machinery, thus blocking the synthesis of long poly(A) tails. Therefore, cellular pre-mRNAs containing short poly(A) tails (around 12 nucleotides) accumulate in the nucleus of IAV-infected 293 cells. The CPSF30 and PABII proteins bind to different sites of the ED of the NS1 protein, suggesting these three proteins might form a ternary complex [51].

Besides cellular proteins, NS1 could also bind to a stem-bulge region in U6 small nuclear RNA (snRNA). U6 snRNA is a component of a spliceosome that catalyzes the excision of introns from pre-mRNA. Through interacting with U6 snRNA, NS1 blocks cellular pre-mRNA splicing [52].

Using HSP70 synthesis as a reporter, it has been shown that IAV inhibits the polyadenylation-site cleavage of the pre-mRNAs. One mutant IAV carrying a temperature-sensitive mutation on NS1 gene failed to inhibit the cleavage at non-permissive temperature, indicating that the NS1 protein is involved in the inhibition of the pre-mRNA cleavage. In this case, IAV causes host shutoff through the down-regulation of cellular mRNA maturation at the point of polyadenylation-site cleavage [53].

NS2 and M2 proteins are encoded from the spliced mRNAs of IAV segment 7 and 8, respectively. NS1 also regulates the splicing of the viral M1 mRNA. Using a proteomics approach, a complex of RNA binding proteins containing NS1-binding protein (NS1-BP) and heterogeneous nuclear ribonucleoproteins (hnRNPs) was found to be involved in the regulation of viral M1 mRNA by NS1. Reduction of NS1-BP and/or hnRNP K modulated M2/M1 mRNA ratios, which resulted in decreasing M2 level and suppressing viral replication [54]. Therefore, viral mRNAs are efficiently processed prior to nuclear export, and thus subject to NS1-mediated inhibition of cellular mRNA nuclear export.

### 2.3. Degradation of Cellular mRNAs by PA-X ((3) in Figure 1)

Early studies showed that cellular mRNAs degradation in IAV-infected COS-1 cells is accelerated [55]. Both cellular mRNAs isolated from cytosolic fractions as well as total mRNAs showed a rapid decay while IAV M1 mRNA levels were below detection limits. In contrast, these cellular mRNAs were stable in un-infected cells. Thus, IAV infection induces cellular mRNA degradation, which is in turn causing host shutoff. An unexpected role of PA-X in cellular RNA degradation was reported by Jagger et al. in 2012 [56]. IAV PA and the PA-X proteins are both encoded from viral segment 3. PA is translated directly from PA mRNA while PA-X is translated as a +1 frameshift product from the same PA mRNA. During the PA mRNA translation, ribosome shifts to a uracil (U)-rich stretch which is followed by a rare codon (UCC_UUU_CGU) [57]. The ribosomal frameshifting occurs rarely (< 2%), but its occurrence can be promoted during the slow rare codon decoding process. The translated product of +1 frameshift is a PA-X protein which consists of the same first N-terminal 191 a.a. of PA, including an endonuclease domain and a unique C-terminus 41 or 61 a.a. (i.e., X-ORF). Most IAV PA-X proteins have an X-ORF of 61 residues. The +1 frameshift motif of IAV segment 3 is highly conserved, implying that it is essential for the viral life cycle [28,58].

With the N-terminal endonuclease domain of PA, PA-X has the ability to degrade RNA [59,60]. PA-X has stronger activity to shut off host genes than full-length PA or the N-terminal domain of PA alone, demonstrating that the unique C-terminal X-ORF is important in suppressing gene expression. Indeed, the N-terminal 15 amino acids, particularly the 6 basic a.a. of the X-ORF, are identified to be important for PA-X’s host shutoff activity [61]. PA-X can target RNAP II-transcribed RNAs in the nucleus, including non-coding RNAs that are not destined to be translated, while sparing products of RNAP I and RNAP III [62].

Target specificity of PA-X should reside in the X-ORF, which is different from that of PA. Proteomic analysis has identified a number of cellular proteins involved in mRNA processing that interact with X-ORF (61 and 41 a.a.). For example, NUDT21, a component of the CFIm complex involved in pre-mRNA polyadenylation, has been shown to interact with PA-X by co-immunoprecipitation. Additionally, cellular pre-mRNA processing proteins (CPSF5/6) that modify the 3′ end of the nascent transcript were also found to interact with PA-X. Thus, this study indicated that PA-X degraded nascent cellular RNAP II transcripts specifically through interacting with RNA splicing machinery [29]. A translated mRNA that does not undergo canonical 3′ end processing was not degraded by PA-X, suggesting that the translation itself is not sufficient to initiate RNA degradation. Moreover, PA-X also degraded the un-translated RNAP II transcripts such as endogenous long non-coding RNAs [62]. The target specificity of PA-X makes it different from other viral host shutoff proteins that target actively translated mRNAs in the cytosol [33]. Linking RNA degradation by PA-X with RNAP II transcription together spared viral mRNAs and genomic vRNAs automatically from PA-X activity, as they are produced by the viral RdRp [62].

Quantitative subcellular localization analysis revealed that PA-X was distributed in both the nucleus and the cytosol. Degradation of cellular transcripts likely occurs in the nucleus, as PA-X is enriched in the nucleus and its nuclear localization correlates with the reduction of target RNA levels. The first nine amino acids in the X-ORF of PA-X (a.a. 192–200) were sufficient for nuclear localization [62], but an additional six residues (a.a. 201–206) were required to induce the maximum host shutoff activity base on the study with intact PA-X. Furthermore, recent studies also found that the a.a. 233–252 in the C-terminal region of PA-X enhances the host shutoff. Thus, both the N- and C-termini of PA-X contribute to its overall shutoff ability [63].

PA-X proteins in general are short-lived with a half-life range from 30 min to 3.5 h depending on viral strains [64]. Sequences in the variable PA-X C-terminal domain are primarily responsible for regulating its half-life. PA-X with a longer half-life has stronger host shutoff activity than that with a shorter half-life. N-terminal acetylation is a major post-translational modification in eukaryotes catalyzed by N-terminal acetyltransferases (NATs), NatA through NatF. The N-terminal Met-Glu residues of PA-X are found to be post-translationally modified by an acetyltransferase, NatB, which comprises NAA20 and NAA25. The host shutoff activity of PA-X was suppressed in NatB-deficient HAP1 cells, and PA-X mutants that are not acetylated by NatB showed reduced shutoff activities. Thus, N-terminal acetylation is required for the host shutoff activity of PA-X [65].

It is unclear whether the two X-ORF variants (61 a.a. and 41 a.a.) have differential activities. It is difficult to compare the domain function of these two variants, as they naturally occur in distinct strains that may also carry other changes in the PA-X sequences (for a review, see [35]).

Afterthe cleavage by PA-X endonucleolytic activity, the complete degradation of cellular RNAP II transcripts becomes dependent on Xrn1, a cellular exonuclease [62].

Using four recombinant influenza A/California/04/2009 (pH1N1) viruses containing mutations affecting the expression of active PA-X and NS1, Chaimayo et al. analyzed the host shutoff activities of these recombinant viruses. Their results showed that PA-X was the major contributor in reducing general cellular protein expression while NS1 specifically targeted host mRNAs related to IFN signaling pathways and cytokine release. Thus, different IAV proteins with host shutoff activity may have different target specificities [66].

### 2.4. Blockage of Cellular mRNA in the Nucleus by NS1 ((4) in Figure 1)

Nuclear export of cellular mRNAs encoding anti-viral proteins is critical for the control of viral infections [67]. IAVs have also evolved multiple ways to inhibit nuclear export of cellular mRNAs [67]. The NS1 of influenza A/WSN/33 H1N1 (WSN) has been demonstrated to form an inhibitory complex with mRNA export factors such as NXF1-NXT1, Rae1, and E1B-AP5 in an RNA-independent manner. These interactions resulted in the blockage of the nuclear export of cellular fully processed mRNAs [68]. Furthermore, the target of NS1 has been identified as the mRNA export receptor NXF1-NXT1. The crystal structure of the NXF1–NXT1–NS1 complex showed that NS1 prevents the interaction between NXF1-NXT1 and nucleoporins, thereby inhibiting mRNA export [69].

On the contrary, NS1 promotes the nuclear export of viral M1 mRNA through interacting with NXF1 [70].

### 2.5. Selective Translation of Viral mRNAs ((5) in Figure 1)

IAVs have developed several possible mechanisms to enhance translation of viral mRNAs (mentioned in the following section). Through enhancing viral mRNAs translation, the cellular mRNAs translation will be reduced indirectly.

As discussed above, different mechanisms for host shutoff were deployed by IAVs, suggesting that re-directing cellular gene expression is important for successful IAV infection. Why are so many different mechanisms required for IAV infection? One possible explanation is its broad host range. IAV can infect a variety of animals, including wild aquatic birds, poultry, and terrestrial mammals. The mechanisms used by different IAV strains from various hosts for host shutoff should not be the same [71]. In agreement with this hypothesis, many of the proposed functions for NS1 and/or PA-X are IAV strain-specific, though the expression of these two viral proteins is conserved. Moreover, host shutoff activity is subject to a strict balance between NS1 and PA-X proteins, which determines the successful IAV propagation [72].

The main function of host shutoff is likely to reduce the expression of immune-related genes, e.g., type I IFNs. Host shutoff may indirectly enhance viral mRNAs translation due to reduced competition from cellular mRNAs for cellular translational machinery [35]. In addition to host shutoff, IAV also utilized other strategies to suppress innate immune responses.

## 3. Suppression of Host Innate Immune Responses

Cellular innate immune responses are induced after IAV infection through various pattern recognition receptors (PRRs), such as RIG-I-like receptors (RLRs) including retinoic acid-inducible gene I (RIG-I) and melanoma differentiation-associated gene 5 (MDA5) [73]. RIG-I and MDA5 contain N-terminal tandem caspase activation and recruitment domains (CARDs) that function in signaling. Activation of RIG-I or MDA5 leads to interactions of the CARDs with the CARD of the mitochondrial anti-viral signaling protein (MAVS, also known as IPS-1 or VISA protein), the essential signaling adapter protein of the RLRs [74]. CARD–CARD interactions initiate a signaling cascade, such as activation of TBK1 and IKKε protein kinases which phosphorylate and activate the transcription factors interferon regulatory factor 3 (IRF3) and NF-κB. Upon activation, IRF3 and NF-κB translocate from the cytosol to the nucleus to turn on the expression of innate immune response genes, including interferons (IFNs) and pro-inflammatory cytokines, such as TNF-α and IL-6 [75,76]. Secreted IFNs bind to IFN receptors in a paracrine and/or autocrine manner, which activates the JAK/STAT signaling pathway to induce the expression of hundreds of IFN-stimulated genes (ISGs) whose products could restrict viral replication [77].

To ensure viral replication, IAVs have developed various mechanisms to suppress the innate immune responses including prevention of viral RNAs detection by RLRs, inhibition of the activities of proteins involved in signaling events of IFN production, shut off of the IFN gene expression, and inhibition of ISG activities, mainly through viral NS1, PB1-F2, and PA-X proteins [63,78,79,80] (Figure 2). PB1-F2 protein encoded from a + 1 ORF of PB1 gene is expressed in many, but not all, IAV strains [26,81]. The full-length PB1-F2 with 90 a.a. is generally produced in avian IAVs, while the PB1-F2 with 78 a.a. is generated in mammalian IAVs due to a premature stop codon [82].

### 3.1. Prevention of the Viral RNAs Detection by RLRs

RIG-I recognizes uncapped 5’-triphosphate RNA present in the genomes of RNA viruses [75]. Previously, Weber et al. demonstrated that encapsulation of the IAV RNAs in the vRNPs prevents the recognition of viral genomic RNAs by RIG-I and blocks its subsequent activation. Moreover, the polymorphism of PB2 at position 627 affects its binding affinity to NP. The reduced binding affinity of PB2 to NP allows better access of RIG-I to the NP-associated viral RNAs [83]. Thus, genome encapsulation allows IAV to avoid RIG-I recognition. Additionally, NS1 could block the recognition of viral 5’-triphosphate RNA by RIG-I through interacting with RIG-I [84].

RIG-I also recognizes 5’-triphosphate dsRNA and MDA5 senses long dsRNAs generated as replicative intermediates for many RNA viruses [85]. In addition to cellular mRNAs degradation, PA-X has also been proposed to function in viral dsRNAs degradation to evade recognition by various PRRs as PA-X mutated IAV induces higher level of IFN than the wild type [86].

### 3.2. Inhibition of Proteins Involved in the Signaling Leading to IFN Production

IAV NS1 reduces the RIG-I expression. A previous study clearly showed that NS1 post-transcriptionally prevents RIG-I pre-mRNA processing by binding to the RIG-I pre-mRNA, primarily the intronic sequences [87]. NS1 also inhibits the RIG-I activation. Two E3 ligases, TRIM25 and Riplet, are responsible for the ubiquitination of RIG-I, which is critical for its activation [88]. It has been shown that NS1 causes attenuated RIG-I ubiquitination and activation by interacting with TRIM25 and Riplet [89].

The mitochondrial outer membrane protein MAVS, a downstream effector of RIG-I, is a central signaling molecule in viral RNA recognition. The mitochondrial membrane potential is essential in RLR signaling and the activation of NLRP3 inflammasome upon IAV infection [90]. PB1-F2 primarily accumulates in mitochondria via the translocase of the outer membrane 40 (Tom40) channels in IAV-infected HEK293 cells [27]. Through interacting with MAVS, PB1-F2 attenuates mitochondrial membrane potential, which suppresses the activation of RLR signaling and in turn reduces IFN production [91]. Additionally, IAV PB2 protein could also interact with MAVS and inhibit the expression of beta IFN [92].

MAVS recruits the IKK family members, such as IKKε and TBK1, and then activates transcription factors including NF-κB and IRF3. Through its ED, NS1 interacts with IKKα/IKKβ and inhibits IKK-mediated NF-κB activation and production of the NF-κB-induced anti-viral genes [93]. Additionally, PB1-F2 could interact with IKKβ, and then block NF-κB binding to its target genes [94]. Finally, Leymaire et al. reported that PB1-F2 also interacts with cellular calcium-binding and coiled-coil domain 2 (CALCOCO2), a TBK1-binding protein, and suppressed IRF3/IRF7 activation signaling induced by TBK1 [95].

A20, a ubiquitin-editing protein, is demonstrated to be a negative regulator of IRF3 signaling [96]. NS1 induces A20 expression and down-regulates RIG-I signaling indirectly [97].

Recently, NS2 was found to interact with IRF7 using co-immunoprecipitation experiments in HEK293 cells or A549 cells. Through this interaction, NS2 blocks the activation of IRF7 and thus suppresses the production of IFN-β [98].

PA-X could inhibit type I IFN production via RIG-I-MAVS pathway though the mechanism is unknown [99].

### 3.3. Shut off of the IFN Gene Expression

As discussed above, IAVs suppress cellular gene expression via various mechanisms mainly through NS1 and PA-X proteins. The expression of IFN genes will be suppressed by these mechanisms.

### 3.4. Inhibition of ISG Activities

Through binding to and/or degradation of dsRNAs, NS1 and PA-X prevent the activation of two ISGs, protein kinase R (PKR) and 2′–5′ oligo A synthetase (OAS)/RNase L pathway [52,100]. Besides dsRNA, PKR can be activated by the protein activator of interferon-induced protein kinase (PACT) [101]. NS1 also prevents the activation of PKR through directly interacting with this protein or through binding to PACT [102].

Interferon-induced transmembrane protein 3 (IFITM3), another ISG, is important for the cells to fight against a variety of RNA viruses, including influenza virus. The expression of IFITM3 was found to be regulated by eukaryotic translation initiation factor 4B (eIF4B), an integral component of the translation initiation apparatus. IAV infection induces lysosomal degradation of eIF4B protein through NS1 protein and thus inhibits IFITM3 expression [103].

### 3.5. Suppression of Inflammasome Formation

In addition to LRLs, some other PRRs such as NACHT, LRR, and PYD domains-containing protein 1 (NLRP1), NLRP3, NLR family CARD domain-containing protein 4, and absent in melanoma 2 will recruit apoptosis-associated speck-like protein (ASC) and caspase-1 to form the inflammasome and initiate anti-viral immune responses to eliminate viral infection [104]. On the other hand, viruses have developed different strategies to prevent inflammasome formation.

It has been shown that IAV NS1 protein interacts with NLRP3 resulting in impaired ASC speck formation and attenuated ASC ubiquitination, which are relevant to inflammasome formation [105,106].

The most obvious role of suppression of the cellular innate immune responses is to avoid the translational inhibition of viral mRNAs by the ISGs, such as PKR. In addition to suppression of the innate immune responses, IAV also deployed other mechanisms to selectively translate its mRNAs.

## 4. Selective Translation of Viral mRNAs

IAVs employ cellular translation machinery to synthesize viral proteins. Regulations of cap-dependent translation of eukaryotic mRNAs mainly occur in the initiation step, i.e., the cap-binding complex [eukaryotic initiation factor 4F (eIF4F) composed of eIF4E, eIF4G, and eIF4A] binding to the 5’ end of mRNA. Then, the 43S preinitiation complex will be recruited by eIF4F to form 48S initiation complex [107]. The 48S then scans along mRNAs to find a start codon. IAV mRNAs contain a short-capped oligonucleotide sequence at the 5′ ends cleaved from the cellular RNA by a “cap-snatching” mechanism, followed by a common viral sequence and the 3′ of these viral mRNAs are polyadenylated. Although cellular and viral mRNAs are structurally similar, IAV infection causes a significant decrease in cellular mRNA translation while viral mRNAs are efficiently translated [108,109]. Additionally, several reports have shown that IAV protein synthesis but not cellular protein synthesis proceeds efficiently upon functional impairment of the cap-binding protein eIF4E [110,111]. In contrast, eIF4A (helicase) and eIF4G (the scaffolding factor) are required for viral mRNA translation [112]. As mentioned earlier, the viral RdRp conducts the cap-snatching, binds selectively to the common viral sequence of viral mRNAs, and serves as a substitute for eIF4E for viral mRNA translation. Thus, transcription by RdRp is required to secure translation of viral mRNAs that are largely eIF4E-independent, perhaps through direct recruitment of the eIF4G initiation factor [113] (Model A in Figure 3).

However, Bier et al. suggests that the viral RdRp might not be in the protein complex with viral mRNAs [114]. Using chimeras containing the non-coding and coding regions of cellular and viral mRNAs, Garfinkel et al. demonstrated that sequences in the 5′-UTR of viral mRNAs, common to all viral mRNAs, played a critical role in directing selective translation of viral mRNAs [115]. Polysome analysis further confirmed that a 45-nucleotide sequence in the 5′-UTR of the IAV NC mRNAs was necessary and sufficient to re-direct the cellular translational machinery from cellular mRNAs to viral mRNAs [115]. Thus, the transacting factors interacting with 5′-UTR of viral mRNAs should be responsible for the selective viral mRNA translation. NS1 protein was shown to bind to the 5′-UTR of viral mRNAs using Northwestern analysis. Interaction of NS1 with a specific region of the 5′-UTR of viral mRNAs was confirmed using gel shift analysis with purified recombinant NS1 [116]. Indeed, co-expression of NS1, but not of NS2 protein, led to increases in the translation of viral mRNAs (e.g., NP or M1) but not control mRNA (e.g., CAT or LacZ). As expected, 5′-UTR of viral mRNAs in part accounts for the effect of selective translation. Moreover, the translational enhancement of viral mRNAs by NS1 protein was due to an increase in the translation initiation rate, since the sizes of NP-specific polysomes, but not those of LacZ-specific polysomes, were significantly higher in COS-1 cells co-expressing NS1 protein than in those expressing only the NP gene [117]. NS1-interacting proteins should play a role in the enhanced translation of viral but not mRNAs. The eIF4G was co-immunoprecipitated with NS1 either in IAV-infected COS-1 cells or in COS-1 cells transfected with NS1 gene. The interaction of NS1 and eIF4G proteins was further demonstrated using affinity chromatography studies and their direct interaction in an RNA-independent manner was proved using a pull-down experiment [118]. When standard immunoprecipitation, biochemical purification and RNA immunoprecipitation assays were performed to investigate the association of viral and cellular proteins with viral mRNA, a previous study found that viral mRNA associates with NS1 and cellular poly(A)-binding protein 1 (PABP1) [114]. Interactions of NS1 and PABP1 were also demonstrated both in vivo and in vitro in an RNA-independent manner [119]. NS1 also interacts with eIF4G, next to its PABP1-interacting domain [119]. Collectively, these previous studies have suggested that NS1 enhances the translation of viral mRNA by binding to conserved sequences in the viral 5′-UTR. The interactions of NS1, PABP1, and eIF4G within the 5′-UTR of viral mRNAs promote the specific recruitment of the 43S to viral mRNAs and allow the preferential translation of viral mRNAs [119] (Model B in Figure 3).

In contrast, a study reported that NS1 does not directly bind to sequences in the viral 5′-UTR indicating that NS1 is not responsible for the specificity to enhance viral mRNA translation [120]. If NS1 does not interact with viral mRNAs, eIF4E is required to recognize and bind to viral mRNAs and then NS1 interacts with eIF4E and eIF4G. Indeed, interaction of NS1 with the 5′-m7G-mRNA-eIF4E-eIF4G complex has been reported [121]. Cruz et al. showed that NS1 can bind this complex with or without the presence of 5′-m7G-mRNA. Interestingly, NS1 can bind to eIF4E only in the absence of 5′-m7G-mRNA. A model was proposed based on this study and previous reports [121]. However, it did not explain how NS1 enhances viral mRNAs translation but not cellular mRNAs translation (Model C in Figure 3).

The 5ʹUTR of IAV mRNAs are highly conserved and rich in adenosine residues. A recent study has shown that the PABP1 binds to the 5′UTR of the viral mRNAs and recruits eIF4G in an eIF4E-independent manner [122]. These results indicated that PABP1 may promote eIF4E-independent translation initiation of viral mRNAs. Based on these results, a model was proposed: PABP1 binds both to the 3′ poly(A) tail and to the 5ʹUTR of viral mRNAs. Then, PABP1 in the viral 5ʹUTR recruits eIF4G, eIF4A, and eIF4B. Finally, the 43S is recruited by eIF4G to initiate translation in an eIF4E-independent manner [122]. In agreement with this model, NS1 of different IAV strains was found to bind as a homodimer to PABP1. However, NS1 only binds to PABP1 free of RNA but not with bound poly(A) RNA. Thus, translation enhancement does not occur by NS1 interacting with the PABP1 molecule attached to the 3′-poly(A) tails of mRNAs. Furthermore, de Rozieres et al. suggested that the function of the NS1–PABP1 complex appears to be distinct from the classical role of PABP1, which binds to the 3′-poly(A) tails of mRNAs [123]. If this is true, the translational enhancement of NS1 on viral mRNAs may be simply due to its anti-IFN functions, e.g., blockage of PKR. It is obvious that PKR inhibits global translation through phosphorylation of eIF2α Indeed, IAV without NS1 could still replicate in Vero cells that do not have type I IFN genes [124], suggesting the role of NS1 in translation is not a requirement for IAV replication (Model D in Figure 3).

In addition to PABP1, other cellular proteins may be also involved in the selective translation of IAV mRNAs. Recently, using a CRISPR/Cas9-knockout selection system designed to search for cellular factors required for viral replication, many cellular factors were identified. Among them, IFIT2, an ISG with anti-viral activity, was repurposed by IAV to promote viral gene expression [125]. IFIT2 directly interacts with viral and cellular mRNAs in AU-rich regions demonstrated by CLIP-seq, with bound cellular transcripts enriched in ISGs. IFIT2 is further revealed to prevent ribosome pausing on bound mRNAs. A model was proposed based on these findings [125]: Before viral infection, IFIT2 enhances translation of cellular mRNAs, including ISGs, to suppress viral infections. After successful IAV infection, IFIT2 is re-purposed to enhance the translation of viral mRNAs through directly binding to IAV mRNAs in AU-rich regions.

It will be interesting to know whether PABPI and/or IFIT2 are involved in the selective translation of mRNAs from other viruses. It is important to find out whether there are more other cellular proteins involved in the selective translation of IAV mRNAs. More studies are needed to address these issues.

## 5. Discussion

IAVs have utilized many mechanisms to shut off cellular gene expression mainly through NS1 or PA-X. As discussed above, both NS1 and PA-X proteins exerted their functions in the nucleus, implying that IAV suppresses cellular gene expression prior to translation in the cytosol. Notably, PA-X protein has been reported to degrade mRNAs both in the nucleus and the cytosol [126]. If this is the case, IAV would suppress the cellular gene expression more thoroughly. Then, the question is how PA-X selects cellular mRNAs but not viral mRNAs for cleavage in the cytosol?

It is well known that NS1 (both transiently and stably expressed) enhances the translation of transient gene expression [127,128]. Recently, a report demonstrated that inducible intact NS1 or only the ED of NS1 attenuates the transcription of transient gene expression [45]. If the same results could be obtained using different ways (transient, stable, or inducible) to express NS1 proteins of different IAV strains in various cell types, more studies are required to determine the regulations of NS1 in these two events.

NS1 suppresses cellular gene expression through various mechanisms including attenuation of transcription, prevention of pre-mRNA maturation, and interference of mRNA nuclear export. Further studies are needed to clarify the spatial and temporal regulations among these mechanisms. Previous studies suggested that a strict balance between NS1 and PA-X proteins in host shutoff action is needed to determine the successful progression of IAV infection [66,72]. Thus, additional studies on the reciprocal regulation between NS1 and PA-X proteins in host shutoff are required. Similarly, more studies on the concerted effect of PA-X, PB1-F2, and NS1 in avoiding the cellular innate immune responses for host adaptation are needed [129].

A recent report indicated that PABP1 bound to the viral 5ʹUTR may promote eIF4E-independent translation initiation (Model D in Figure 3) [122]. In this model, NS1 did not interact with viral 5′UTR directly and its enhanced translation of viral mRNAs demonstrated in other studies may be just due to its blockage of PKR. If translational enhancement of viral mRNAs by NS1 is due to its blockage of PKR, this model did not explain the disagreements from previous studies: the N-terminal RBD of NS1 is responsible for the viral translational enhancement while the C-terminal ED interacts and inhibits PKR [130,131]. Further studies are required to clarify this discrepancy.

## 6. Conclusions

Over the past three decades, much has been learned about the strategies utilized by IAV to ensure sufficient translation of viral mRNAs. These strategies include host gene shutoff, suppression of host innate immune responses, and selective translation of viral mRNAs, of which the detailed molecular mechanisms are yet to be elucidated.

## Figures and Tables

**Figure 2 pathogens-11-01521-f002:**
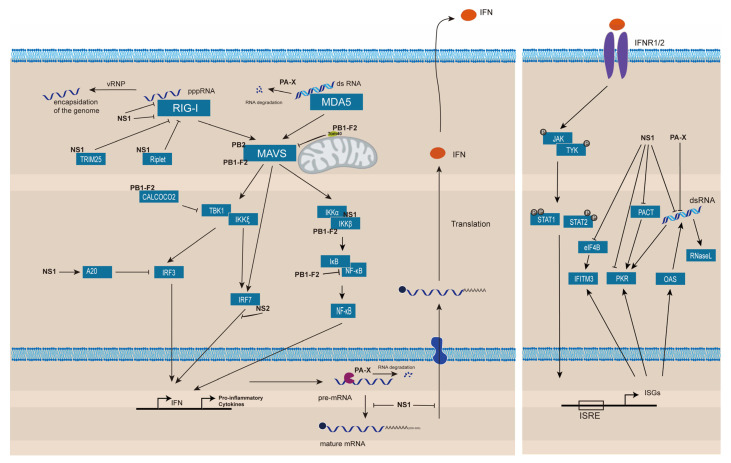
Various mechanisms deployed by IAV to suppress innate immune responses. Following IAV infection, 5ʹ-triphosphate RNAs will be recognized by RIG-I and dsRNAs will be recognized by MDA5, resulting in induction of the innate immune response, such as type I IFNs. IFN induction is regulated by various transcription factors, such as the IFN regulatory factors 3 and 7 (IRF3/IRF7) and nuclear factor κB (NF-κB). IAV prevents the detection of viral RNAs by RLRs through encapsulation of viral genomes. Moreover, IAV NS1 and/or PB1-F2 inhibit the protein activities involved in signaling events of IFN production. IAV also shuts off the expression of IFN gene and blocks the ISG activities through NS1 and/or PB-X. IFN: interferon; IFNR: IFN receptor; ISRE (interferon-stimulated-response element) represents the region in the promoters of ISEs responsive to IFN induction; IFITM3: interferon-induced transmembrane protein 3; eIF4B: eukaryotic translation initiation factor 4B.

**Figure 3 pathogens-11-01521-f003:**
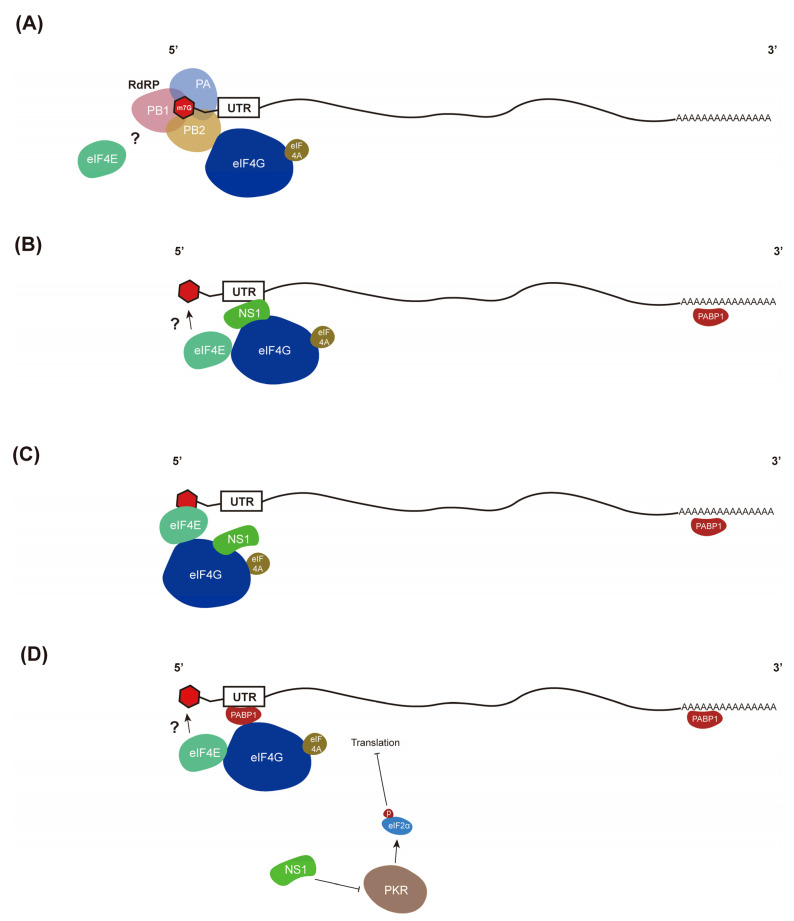
Several models were proposed for the initiation of translation of IAV mRNAs. (**A**) After cap-snatching, PB2 interacts with eIF4G and recruits other factors and 43S pre-translation initiation complex. (**B**) Interactions of NS1 with 5′-UTR of viral mRNAs recruits PABP1 and eIF4G. (**C**) Neither NS1 nor eIF4G interacts with 5′-UTR of viral mRNAs directly; NS1 facilitates translation by binding to eIF4G and recruiting 43S to mRNAs. (**D**) Interactions of PABP1 with the 5′-UTR of viral mRNAs rich in adenosine residues. NS1 enhances translation through blocking the PKR activity but does not interact with viral mRNAs directly.

## Data Availability

Not applicable.

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
