# Peer review of "Strategies of Influenza A Virus to Ensure the Translation of Viral mRNAs"

_pathogens, 2022, doi:10.3390/pathogens11121521_

Round 1

Reviewer 1 Report

This manuscript reviewed the previously published articles and summarized the strategies influenza A viruses (IAVs) takes to ensure sufficient translation of viral mRNA, mainly focusing on host gene shutoff, suppression of host innate immune responses, and four models of selective translation of viral mRNAs. 

There are several points that need attention, as follow.

1. The Introduction gave a summary of the 17 proteins IAV RNA encodes, including their locations and functions. It would be better if there could be a summary figure representing these. 

2. It would be better Figures 1 and 2 could be referenced in the text, pointing to the specific panel. For example, refer to Figure 1(1) in Line 92, refer to Figure 1(2) in Line 112, etc.

3. Lines 245-248. It was mentioned that PA-X was distributed equally in both cytoplasm and nucleus, but then in the next sentence that PA-X is enriched in the nucleus. This is confusing.

4. Figure 1(2)(3): Why are some of the cellular mRNAs blue while some are purple?

5. Figure 1(3): It should be PA-X instead of PAX.

6. In normal eukaryotic translation, apart from eIF4A, eIF4E and eIF4G, eIF4B is also a critical protein. In IAV infection, there has been studies showing that eIF4B plays an important role in host defense and that the viral NS1 protein could induce the lysosomal degradation of eIF4B, as a strategy to overcome host innate immunity by downregulating eIF4B protein (please one example reference article below). This was neither mentioned in the text nor reflected in Figure 3. 

(Example reference article: https://www.ncbi.nlm.nih.gov/pmc/articles/PMC4135930/)

Author Response

This manuscript reviewed the previously published articles and summarized the strategies influenza A viruses (IAVs) takes to ensure sufficient translation of viral mRNA, mainly focusing on host gene shutoff, suppression of host innate immune responses, and four models of selective translation of viral mRNAs.

There are several points that need attention, as follow.

  1. The Introduction gave a summary of the 17 proteins IAV RNA encodes, including their locations and functions. It would be better if there could be a summary figure representing these.

Response: Thanks for the suggestion! Table 1 is added as suggested in lines 75-78.

  1. It would be better Figures 1 and 2 could be referenced in the text, pointing to the specific panel. For example, refer to Figure 1(1) in Line 92, refer to Figure 1(2) in Line 112, etc.

Response: Thanks for the suggestion! Figure 1(1)-(5) has been added in the text as suggested (marked in red in the revised manuscript).

  1. Lines 245-248. It was mentioned that PA-X was distributed equally in both cytoplasm and nucleus, but then in the next sentence that PA-X is enriched in the nucleus. This is confusing.

Response: Thanks for the suggestion! The sentence has been revised in lines 251-252 marked in red.

  1. Figure 1(2)(3): Why are some of the cellular mRNAs blue while some are purple?

Response: blue is for intron while purple is for exon. After splicing, only exons (purple) remain.

  1. Figure 1(3): It should be PA-X instead of PAX.

Response: Thanks for the suggestion! It is corrected as suggested.

  1. In normal eukaryotic translation, apart from eIF4A, eIF4E and eIF4G, eIF4B is also a critical protein. In IAV infection, there has been studies showing that eIF4B plays an important role in host defense and that the viral NS1 protein could induce the lysosomal degradation of eIF4B, as a strategy to overcome host innate immunity by downregulating eIF4B protein (please one example reference article below). This was neither mentioned in the text nor reflected in Figure 3.

(Example reference article: https://www.ncbi.nlm.nih.gov/pmc/articles/PMC4135930/)

Response: Thanks for the suggestion! I must have lost this reference. This reference has been added in lines 408-413 (marked in red) and Fig. 2.

Reviewer 2 Report

This is a well written review manuscript which I haven’t found solid mistake. The Comments below are just my personal opinions and suggestions that the authors should feel free to take it or not.

1. It is well established that the main function of NS2 is to take part in the translocation of viral proteins, however, there are some researches showing that NS2 may lead to negative regulation of type I interferon. It could be included in this review.

2. I find section 2(5) is a little awkward to be presented there, for “selective translation” dose not fit in “gene shutoff” and section 4 would discuss it thoroughly. Even if the authors feel it is necessary to bring up this topic, I did not see the point to elaborate it using so many words.

3. Same problem with section 3(3).

Author Response

This is a well written review manuscript which I haven’t found solid mistake. The Comments below are just my personal opinions and suggestions that the authors should feel free to take it or not.

  1. It is well established that the main function of NS2 is to take part in the translocation of viral proteins, however, there are some researches showing that NS2 may lead to negative regulation of type I interferon. It could be included in this review.

Response: Thanks for the suggestion! This reference has been added in lines 393-395 (marked in red) and Fig. 2. Also ref. 11, 22 and Table 1.

  1. I find section 2(5) is a little awkward to be presented there, for “selective translation” dose not fit in “gene shutoff” and section 4 would discuss it thoroughly. Even if the authors feel it is necessary to bring up this topic, I did not see the point to elaborate it using so many words.

Response: Thanks for the suggestion! Only two sentences are in section 2(5).

  1. Same problem with section 3(3).

Response: I do admit that this is redundant. To repeat it just wants to show the reciprocal interaction between these different strategies of IAV.

Reviewer 3 Report

In this review, Li et al. summarize and describe how influenza A viruses (IAVs) divert the cellular machinery to ensure sufficient translation of viral mRNAs inside their host cells. In general, this review is very exhaustive and the figures greatly help to understand the topics.

Overall the writing, phrasing and grammar of the manuscript are excellent and understandable. Very well done. Only a some corrections and rephrasing have to be made in the manuscript itself (see minor points for the text).

If the points mentioned in detail below can be addressed by the authors in a revision, this already excellent review is ready for publication and will be a great contribution the field. Although there are only small corrections, it still is a lot, so I will demand a major revision to see the improvement of this article before publication (but basically it is a minor revision).This article represents high quality scientific writing. Very looking forward to future articles about the topic.

Author Response

In this review, Li et al. summarize and describe how influenza A viruses (IAVs) divert the cellular machinery to ensure sufficient translation of viral mRNAs inside their host cells. In general, this review is very exhaustive and the figures greatly help to understand the topics.

Overall the writing, phrasing and grammar of the manuscript are excellent and understandable. Very well done. Only a some corrections and rephrasing have to be made in the manuscript itself (see minor points for the text).

If the points mentioned in detail below can be addressed by the authors in a revision, this already excellent review is ready for publication and will be a great contribution the field. Although there are only small corrections, it still is a lot, so I will demand a major revision to see the improvement of this article before publication (but basically it is a minor revision).This article represents high quality scientific writing. Very looking forward to future articles about the topic.

Response: Thank you very much for your suggestions! I have revised the manuscript based on your suggestions (marked in the red color in the revised manuscript) except one [In line 447: I think you meant “Northern Blot”]. Actually, [Northwestern analysis] is for the protein-RNA interaction. I do appreciate your help.

Round 2

Reviewer 3 Report

The authors adressed all of my suggestions. Thank you very much. This article is ready for puplication.